# Predictive factors for long-term survival after surgery for pancreatic ductal adenocarcinoma: Making a case for standardized reporting of the resection margin using certified cancer center data

**Dirk Weyhe**[1], **Dennis Obonyo**[1], **Verena Nicole Uslar**[1]*, **Ingo Stricker**[2], **Andrea Tannapfel**[2]

**1** University Hospital for Visceral Surgery, Pius-Hospital, Carl von Ossietzky University Oldenburg, Oldenburg, Germany, **2** Institute for Pathology, Ruhr University Bochum, Bochum, Germany

* verena.uslar@uol.de

**Data Availability Statement:** The data underlying the results presented in this study contain potentially identifying participant information and

## Abstract

Factors for overall survival after pancreatic ductal adenocarcinoma (PDAC) seem to be nodal status, chemotherapy administration, UICC staging, and resection margin. However, there is no consensus on the definition for tumor free resection margin. Therefore, univariate OS as well as multivariate long-term survival using cancer center data was analyzed with regards to two different resection margin definitions. Ninety-five patients met inclusion criteria (pancreatic head PDAC, R0/R1, no 30 days mortality). OS was analyzed in univariate analysis with respect to R-status, CRM (circumferential resection margin; positive: ≤1mm; negative: >1mm), nodal status, and chemotherapy administration. Long-term survival >36 months was modelled using multivariate logistic regression instead of Cox regression because the distribution function of the dependent data violated the requirements for the application of this test. Significant differences in OS were found regarding the R status (Median OS and 95%CI for R0: 29.8 months, 22.3–37.4; R1: 15.9 months, 9.2–22.7; p = 0.005), nodal status (pN0 = 34.7, 10.4–59.0; pN1 = 17.1, 11.5–22.8; p = 0.003), and chemotherapy (with CTx: 26.7, 20.4–33.0; without CTx: 9.7, 5.2–14.1; p < .001). OS according to CRM status differed on a clinically relevant level by about 12 months (CRM positive: 17.2 months, 11.5–23.0; CRM negative: 29.8 months, 18.6–41.1; p = 0.126). A multivariate model containing chemotherapy, nodal status, and CRM explained long-term survival (p = 0.008; correct prediction >70%). Chemotherapy, nodal status and resection margin according to UICC R status are univariate factors for OS after PDAC. In contrast, long-term survival seems to depend on wider resection margins than those used in UICC R classification. Therefore, standardized histopathological reporting (including resection margin size) should be agreed upon.

cannot be shared publicly. The data are available upon request from the Medical Ethics Committee of the Carl von Ossietzky University Oldenburg. Contact: Carl von Ossietzky Universität Oldenburg Fakultät VI Medizin und Gesundheitswissenschaften Medizinische Ethik-Kommission Ammerländer Heerstr. 114-118 26129 Oldenburg Phone: +49 (0) 441 798-3109 Email: med.ethikkommission@uol.d.

**Funding:** The authors received no specific funding for this work.

**Competing interests:** The authors have declared that no competing interests exist.

## Introduction

The prognosis of pancreatic ductal adenocarcinoma (PDAC) is poor despite improvements in surgery and multimodal concepts [1–4]. Even successful surgical resection yields 5-year survival rates of only around 7–25% [5–7]. After adjuvant chemotherapy (CTx) the overall survival (OS) still remains poor [8], though large randomized trials have shown a significant improvement of OS and DFS [9–11]. Generally microscopic involvements of a resection margin (RM) by tumor, perineural invasion, lymphovascular invasion as well as lymph node ratio (LNR) are associated with a poor prognosis [12–19]. RM involvement and the presence of microscopic tumors at time of resection might be the reason for the high rate of local recurrence in over 60% of all patients with PDAC [20–22]. Therefore, in the last decade there has been a debate to redefine resection margin (RM), or rather introduce circumferential resection margin (CRM) consistent to rectal cancer [3, 21, 23–28]. Because of the various proposed definitions, comparison of different studies gets problematic as much confusion exists regarding the exact definition CRM in pancreatic cancer.

According to the definition of the Union for International Cancer Control (UICC), R0 is classified as the absence of tumor cells at the definite resection margin. Contrariwise, many centers in Europe report a residual tumor (R1) according to Royal College of Pathologist (RCP) guidelines whenever tumor cells are present at or within 1 mm of the resection margin [28]. The CRM rule was adopted from rectal cancer surgery, which showed a strong correlation between local or distant recurrence and margin clearance of 1 mm or less (CRM positive) versus a margin clearance larger than 1 mm (CRM negative, S1 Fig) [26, 29, 30]. Controversial data exist on whether tumor involvement in different circumferential resection margins after pancreaticoduodenectomy (PD) differently influences the oncologic outcome of patients suffering from PDAC [3, 19].

Therefore, based on the prospective data of a certified pancreatic cancer center aggregated over 9 years and with at least one-year follow-up, we aim to identify the main variables influencing OS on the one hand, and long-term survival on the other hand in patients with PDAC. For univariate analysis OS was calculated based on the RCP guideline (CRM positive vs. CRM negative), R status based on UICC, nodal status, and the administration of adjuvant chemotherapy to be able to compare our data with already published data. For the main research question, namely multivariate analysis of influences on long-term survival (i.e, 36 months), logistic regression models were established. The aim of this exploratory, multivariate analysis was to determine a multivariate model which best predicts long-term survival in patients with PDAC. In contrast to prior studies, we used multivariate logistic regression, which in contrast to methods like Cox regression is less sensitive to the data structure of the dependent data, and focused explicitly on long-term survival.

## Materials and methods

### Study design and patients

The study was approved by the medical Committee for Research Ethics at the University of Oldenburg (reference number 2019–071), and was registered in the German Clinical Trials Registry (reference number DRKS0017425). It followed the Helsinki Declaration. The need for informed consent is waived by our ethics committee for retrospective studies. For this study, we screened all 463 patients of the certified pancreatic cancer center of the Department of General and Visceral Surgery, Pius Hospital Oldenburg, University of Oldenburg for eligibility (see also Fig 1). All 233 patients who underwent surgery for pancreatic cancer between January 2010 and December 2018 were chosen from this prospectively maintained database. All the

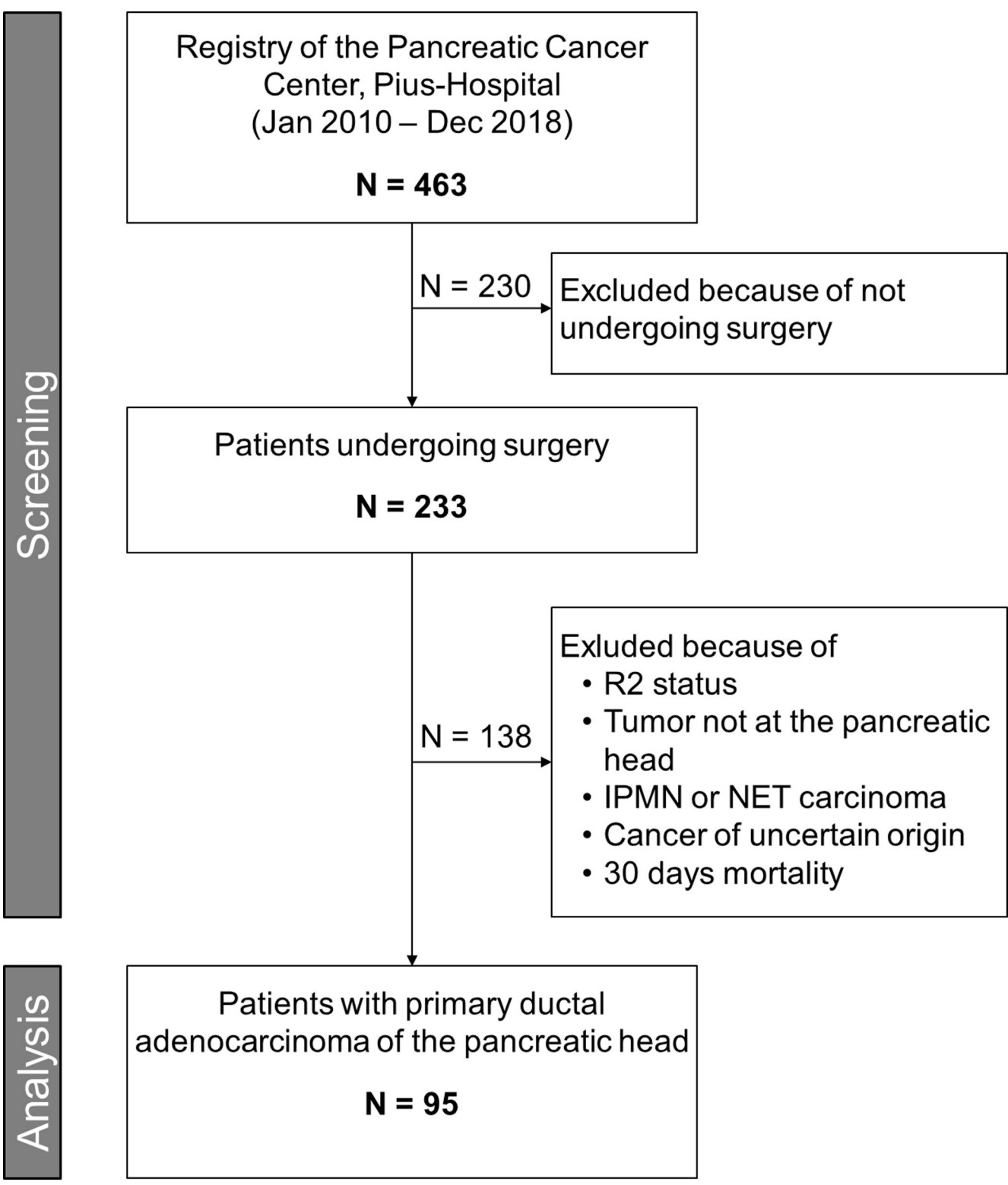

**Fig 1. Study flow chart.**

periampullary carcinomas, bile duct carcinomas, body/tail carcinomas, adenocarcinomas arising in the presence of intraductal papillary mucinous neoplasms (IPMN), neuroendocrine carcinomas and cancers of uncertain origin were excluded. All R2 resection were excluded. Also excepted from the study were patients who died within 30 days after resection. Thus, 95 patients with primary ductal adenocarcinoma of the pancreatic head with macroscopically free margins using the UICC classification R0 or R1 resections; (R0: no tumor cells within the resection margin and R1: tumor cells found in the resection margin without macroscopic residual tumor in situ) were included in the analysis.

Adjuvant therapy was recommended for all patients after curative resection (R0 and R1, UICC 8th edition stage I–III). Chemotherapy with Gemcitabine or FOLFIRINOX (5-Fluorouracil, Irinotecan und Oxaliplatin) was the standard for adjuvant treatment. Neoadjuvant Therapy was administered in 5 patients. Alternative regimens including radio-chemotherapy were used in the setting of clinical trials, e.g. CONKO-007 trial for n = 3 patients.

## Surgical procedures

Patients with PDAC of the pancreatic head underwent pylorus-preserving pancreatoduodenectomy (PPPD) or Kausch-Whipple pancreatoduodenectomy (PD = 1). A pylorus-preserving PD and not a standard Kausch-Whipple PD was the first choice of procedure if the tumor was macroscopically resectable. Segmental or tangential portal or superior mesenteric vein resection was performed if necessary. Prior unknown singular metastases in the liver or in the colic mesentery were resected if a R0-situation was feasible. Lymph node dissection was made in the region of the celiac trunk, hepatic artery and the right aspect of the superior mesenteric artery, and vein as well as retropancreatic tissue. Extended lymphadenectomy was not performed [31, 32]. Intraoperative frozen sections from the bile duct and pancreatic neck resection margins were performed regardless of the macroscopic free-tumor involvement, and if necessary a re-resection was performed and again examined until tumor clearance was microscopically achieved.

## Histopathological assessment of specimens

For purpose of comparability the histopathological examination was performed as defined by general rules of the UICC classification system (8th Edition of the UICC TNM classification of malignant tumors, 2017). This examination is a standardized procedure performed for all specimen resected in the Clinic of General and Visceral Surgery of the Pius-Hospital by the institute for Pathology in Bochum. They classified the resection margin of each specimen according to the suggestion of the RCP (S1 Fig) [28], which defines R1 resection as tumor cell infiltration at or within 1 mm of the resection margin, as well as based on UICC guideline, which defines R0 as no tumor cells being present at any of the resection margins (including bile duct and pancreatic body/ neck non-inked transection margins), and R1 as tumor cells being present at the inked margin without macroscopic residual tumor. These classifications were used by our tumor documenters and the pathologists, and both are two very common. Therefore, all analyses focus on these two classifications. The tumor stage was determined using the current UICC TNM classification system, 8th edition.

## Clinicopathological data

All data including sex, histology, lymph node metastasis, and tumor type and tumor stage were obtained from the clinical and pathologic records. Patients whose death was clearly documented as attributable to pancreatic cancer were considered to have died of that disease; other deaths were not considered to have been caused by pancreatic cancer. Clinical follow-up data

were obtained by reviewing the hospital records and by direct communication with the attending physicians in a standardized and structured manner based on the data sheet for pancreatic cancer centers of the German Cancer Society (Deutsche Krebsgesellschaft, DKG) by the responsible tumor documentaries, as is mandatory for certified pancreatic cancer centers. Overall survival was calculated from the date of surgical resection of the tumor to the date of death or last follow-up.

### Statistical analysis

IBM SPSS Statistics 25 for Windows was used for statistical analysis. OS was analyzed by the Kaplan-Meier method, and the log-rank test was used for univariate analysis, for later comparison with data from other studies. Cox regression was not appropriate for our data, as the requirements for the application of this statistical test were violated. Therefore, to evaluate which factors influence long-term survival (i.e., > 36 months), we performed a logistic regression analysis after variables were assessed for collinearity and interaction with stepwise backward and forward selection for multivariate analysis, thus arriving at the best model fit. For input and output variables, see results section. Statistical significance was determined as $p < 0.05$ for all tests, with a Benjamini-Hochberg correction for multiple testing for the log-rank tests.

## Results

### Univariate analysis of overall survival

In total, 95 patients were enrolled. The mean follow-up time for OS was 21.8 months with 34% of all data censored. Table 1 shows the clinicopathological characteristics of all patients, as well as sorted by CRM status, and by R status. Cause of death is known for only about 1/3 of all patients. However, in those cases, 90% died because of recurring disease.

**CRM-status.** OS differs on a clinically relevant level with median OS for CRM positive patients (n = 63; 66.3%) being 17.2 months (95% Confidence Interval: 11.5–23.0 months), and median OS for CRM negative patients 29.8 months (95% CI: 18.6–41.1; n = 32; see also Fig 2). This difference is not statistically significant (Log-Rank Test: test statistic (1) = 2.343, p = .126).

When plotting the OS in months of all deceased patients against the size of the CRM in mm (Fig 3), we can observe a tendency towards increasing OS with increasing resection margin. However, because of the sparse data for large resection margins (> 10 mm), no valid statement can be made, based on our data.

**UICC R-status.** Using the current UICC R-classification n = 72 (75.8%) patients had R0, n = 23 R1. Fig 4 depicts Kaplan-Meyer curves for OS according to current R-classification. There was a significant difference in OS between both groups (test statistic (1) = 8.048; p = 0.005). Median OS for the R0 patients was 29.8 months (95% CI: 22.3–37.4), compared with R1 patients' median OS of 15.9 months (95% CI R1: 9.2–22.7).

**N-status.** Of the n = 95 patients, n = 72 (75,8%) had lymph node metastasis (pN1/pN2 as shown in Table 1. The mean lymph node yield was 24 (range: 10–52). There was a significant difference in OS with regard to N-status (Fig 5; median OS and 95% CI in months: pN0 = 34.7, 10.4–59.0; pN1 = 17.1, 11.5–22.8; Log-Rank test statistic (1) = 8.803; p = 0.003).

**Adjuvant chemotherapy.** N = 66 (69.5%) patients received CTx. N = 24 (25.3) did not receive CTx or did not finish the regime, and for n = 5 (5.3%) patients no data was available regarding their post-operative treatment. As depicted in the Kaplan-Meier curves (Fig 6), patients who received CTx showed higher OS compared to those who did not receive or

**Table 1. Patient characteristics for all patients stratified by CRM status and R status respectively.**

| | All patients (n = 95) | CRM negative (n = 32) | CRM positive (n = 63) | R0 (n = 72) | R1 (n = 23) |
|---|---|---|---|---|---|
| Age (mean ± SD) | 67.5 ± 9.8 | 66.5 ± 10.4 | 67.9 ± 9.7 | 68.6 ± 9.2 | 66.2 ± 9.9 |
| Sex (m/f) | 54/51 | 18/13 | 36/38 | 18/13 | 36/38 |
| ASA score | | | | | |
| II (n) | 31 | 8 | 23 | 21 | 5 |
| III (n) | 73 | 23 | 50 | 51 | 15 |
| IV (n) | 1 | 0 | 1 | 0 | 1 |
| pUICC | | | | | |
| IA (n) | 5 | 3 | 2 | 4 | 1 |
| IB (n) | 8 | 4 | 4 | 8 | 0 |
| IIA (n) | 11 | 6 | 5 | 10 | 1 |
| IIB (n) | 59 | 15 | 44 | 44 | 15 |
| III (n) | 9 | 3 | 6 | 6 | 3 |
| IV (n) | 3 | 1 | 2 | 0 | 3 |
| pT | | | | | |
| 1 (n) | 7 | 4 | 3 | 6 | 1 |
| 2 (n) | 15 | 7 | 8 | 15 | 0 |
| 3 (n) | 72 | 21 | 51 | 51 | 21 |
| 4 (n) | 1 | 0 | 1 | 0 | 1 |
| pN | | | | | |
| 0 (n) | 23 | 13 | 10 | 13 | 10 |
| 1 (n) | 63 | 16 | 47 | 16 | 47 |
| 2 (n) | 9 | 3 | 6 | 3 | 6 |
| pM | | | | | |
| 0 (n) | 92 | 31 | 61 | 72 | 20 |
| 1 (n) | 3* | 1 | 2 | 0 | 3 |
| LNR (mean ± SD) | 0.18 ± 0.2 | 0.1 ± 0.14 | 0.22 ± 0.22 | 0.18 ± 0.21 | 0.20 ± 0.18 |
| pG | | | | | |
| G1 | 4 | 3 | 1 | 4 | 0 |
| G2 | 55 | 23 | 32 | 43 | 12 |
| G3 | 33 | 6 | 27 | 23 | 10 |
| no information available | 3 | 0 | 3 | 2 | 1 |
| Tumor size (mean cm$^3$ ± SE) | 13.3 ± 1.4 | 11.4 ± 2.5 | 14.3 ± 1.6 | 12.3 ± 1.5 | 16.5 ± 2.8 |
| Revision 30d (n) | 11 | 5 | 6 | 8 | 3 |
| Systemic therapy | | | | | |
| not recommended (n) | 9 | 2 | 7 | 7 | 2 |
| carried out (n) | 66 | 20 | 46 | 47 | 19 |
| not carried out/aborted (n) | 15 | 7 | 8 | 13 | 2 |
| 3 yrs. survival rate (percentage ± SE) | 32 ± 6 | 38 ± 10 | 29 ± 6 | 40 ± 6 | 12 ± 7 |
| RFS (mean months ± SE) | 39.0 ± 5.3 | 44.2 ± 8.9 | 22.5 ± 2.4 | 45.3 ± 6.2 | 12.2 ± 5.3 |

*Three solitary intraoperatively detected liver metastasis were resected. This minor liver resections had no effect on mortality and survival rate compared to standard pancreaticoduodenectomy

CRM: Circumferential Resection Margin

LNR: Lymph Node Ratio

RFS: Recurrence free survival

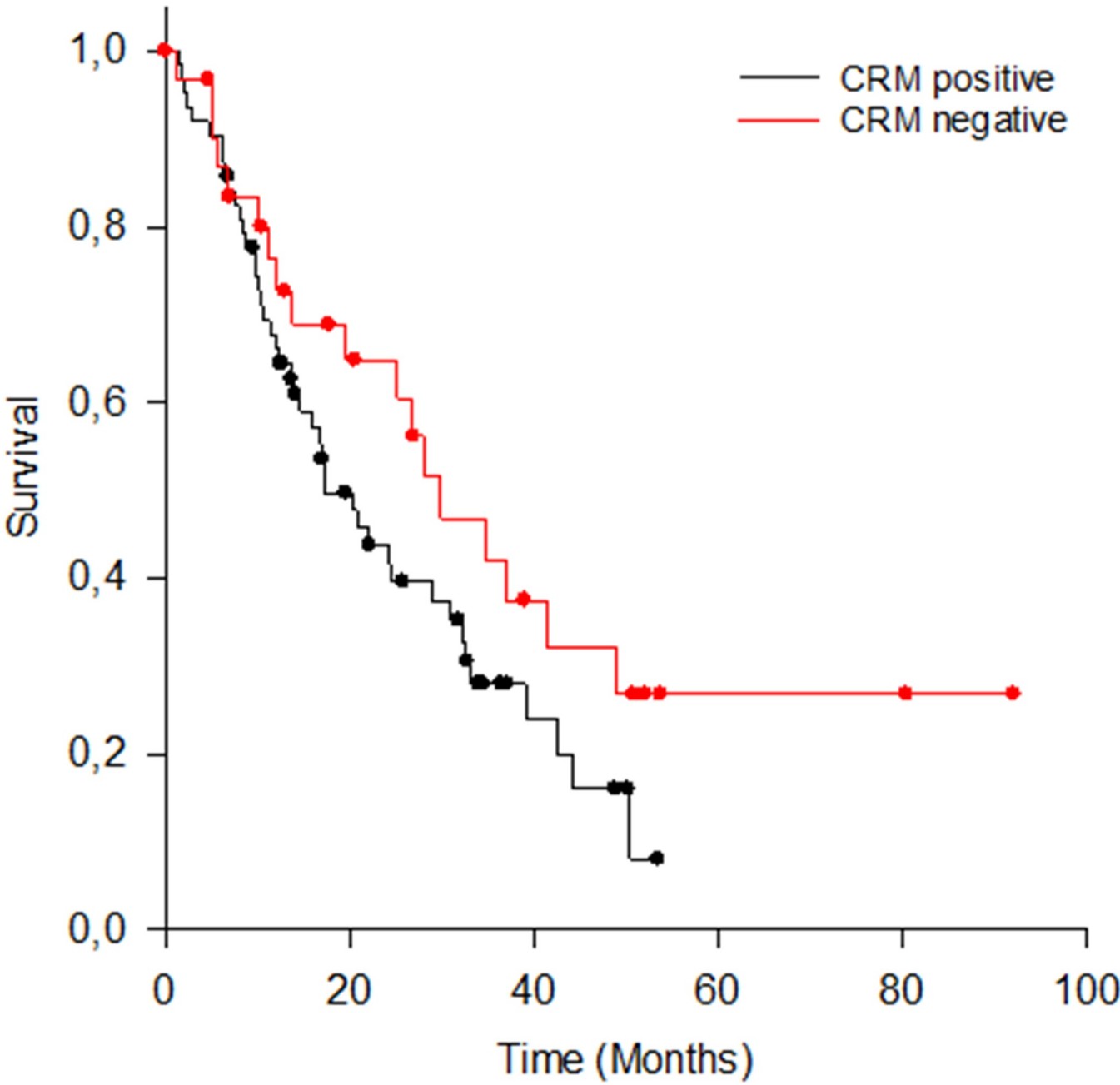

**Fig 2. OS for CRM positive and CRM negative patients.**

complete CTx (OS with CTx = 26.7, 20.4–33.0; without CTx = 9.7, 5.2–14.1; test statistic (1) = 12.751; p < .001).

## Multivariate analysis of long-term survival

Since long-term survival is rather poor in PDAC, one aim of this study was to determine the impact of various variables on long term OS. A logistic regression with iterative backward and forward testing was employed with the following variables as input: age at the time of surgery, sex, ASA score, pT, pN, pM, lymph node ratio, pR, CRM, and systemic therapy

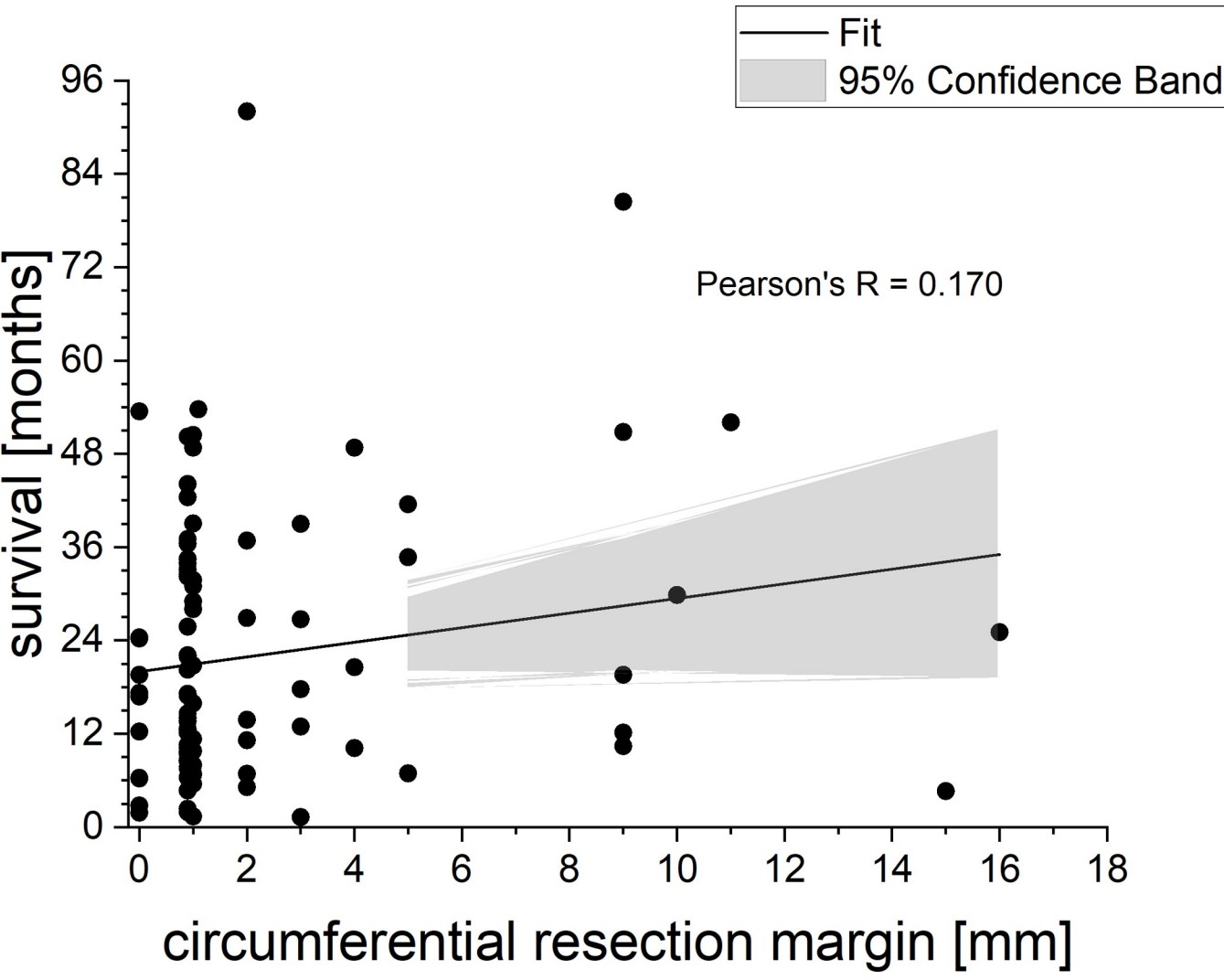

**Fig 3. OS for deceased patients as a function of CRM.**

received or not. On the one hand, variable selection was made for pragmatic reasons, since those variables were readily and more importantly reliably available. On the other hand, the included variables all made sense from a clinical point of view, and represent the most important patient characteristics which might confound the findings. Observed outcome variable was patient survival after more than 36 months (yes or no). Thirty-six months were chosen because it corresponds roughly to the time when about 1/3 of all patients were still alive. We deemed it interesting to analyze why this collective was so long-living as compared to the average PDAC patient. A model with systemic therapy, LNR, and CRM was established which can predict death after 36 months correctly in 70% of all cases and predicts survival longer than 36 months at 71% correct (Likelihood Ratio Test Statistic: 11.749 (p = .008)). However, no single variable was significant by itself (Table 2). This effect is also visible when stratifying OS by CTx, pN status, and CRM status (Fig 7). The R status was not relevant for the prediction of long-term survival, since any model containing R status yielded lower predictability.

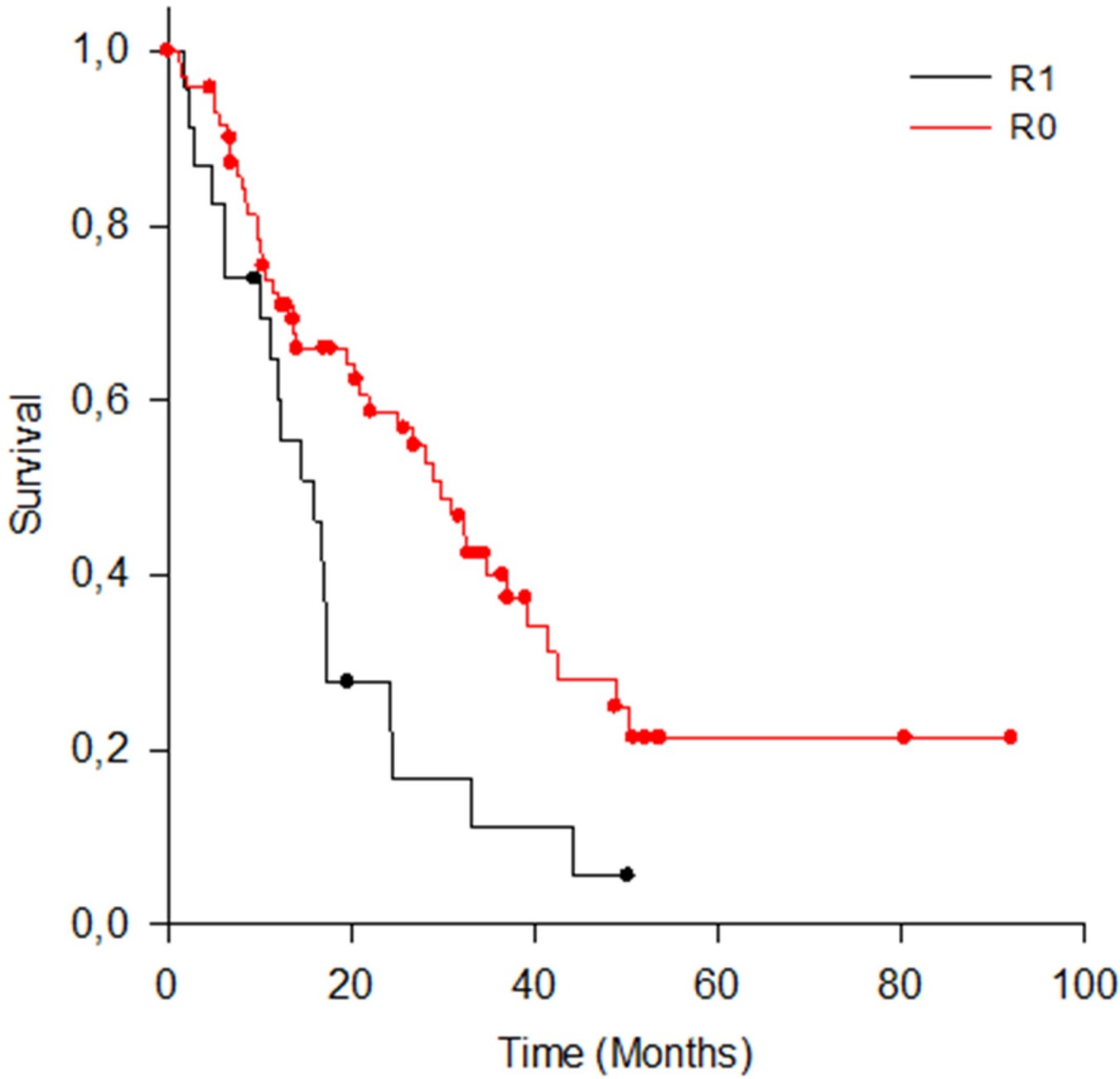

**Fig 4. OS for R1 and R0 patients.**

## Discussion

Pancreatic cancer has one of the shortest rates of overall survival even after successful surgery [2, 8]. Other studies have shown that tumor type, resection margin status, lymph node involvement, tumor stage, vascular invasion and age > 65 years are factors that affect survival rate of patients [8, 13]. However, there are also studies that could not show a significant correlation between nodal status and survival rate. Murakami et al suggested that the prognostic significance of lymph node ratio may depend on the total number of examined lymph nodes [33]. In

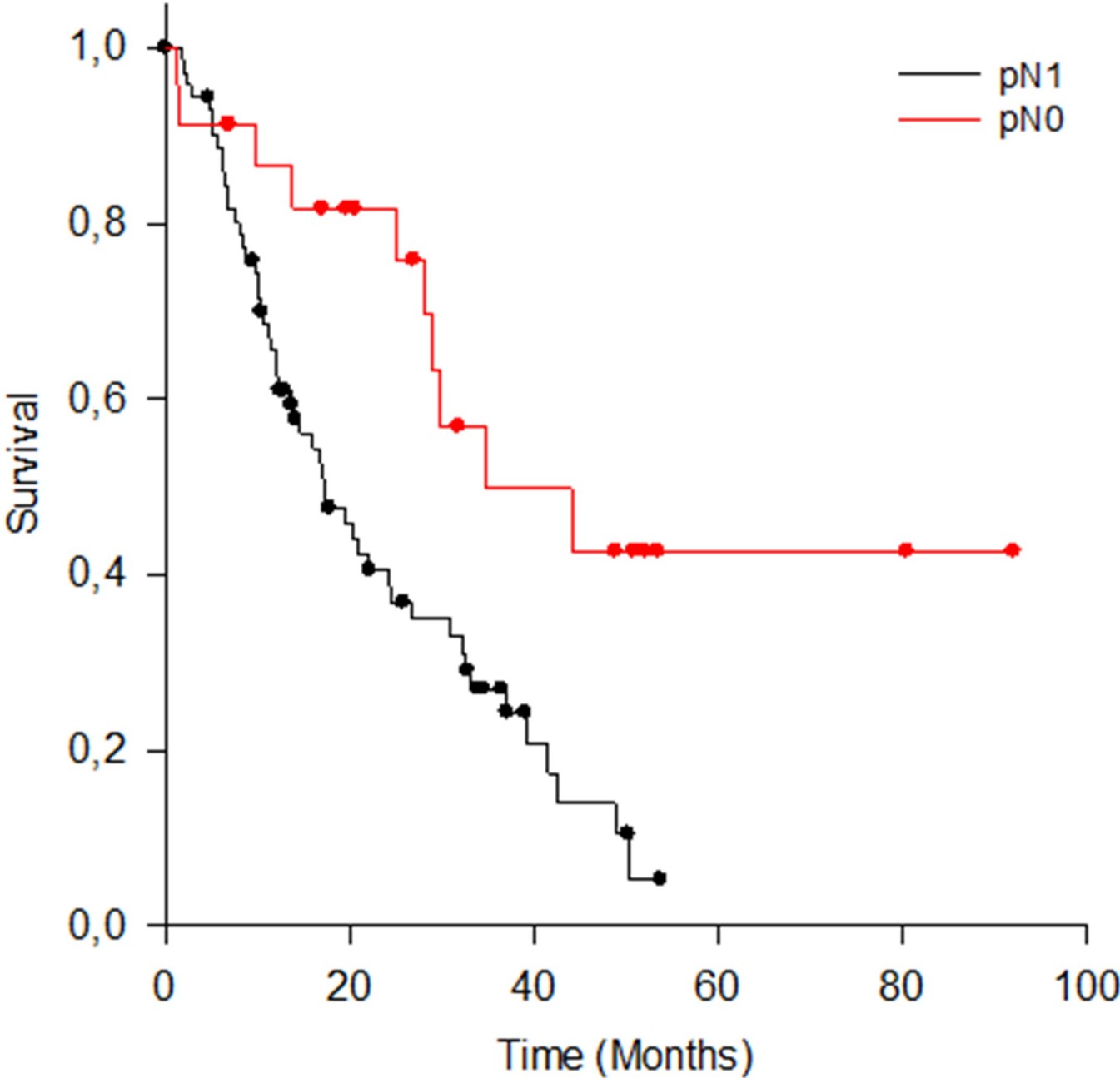

**Fig 5. OS für pN0 and pN1 patients.**

the more common univariate analysis of OS as well as in the special multivariate analysis applied specifically in this study, the presence of metastatic nodal involvement was a significant predictor for OS and long-term survival, respectively. Therefore, nodal involvement remains in our opinion a significant prognostic factor of overall survival, especially in the long term.

Although resection margin involvement is an established prognostic factor for PDAC in several studies, heterogenous histopathological reporting makes it almost impossible to compare the impact of margin status on patients' outcome. This could be one explanation for the

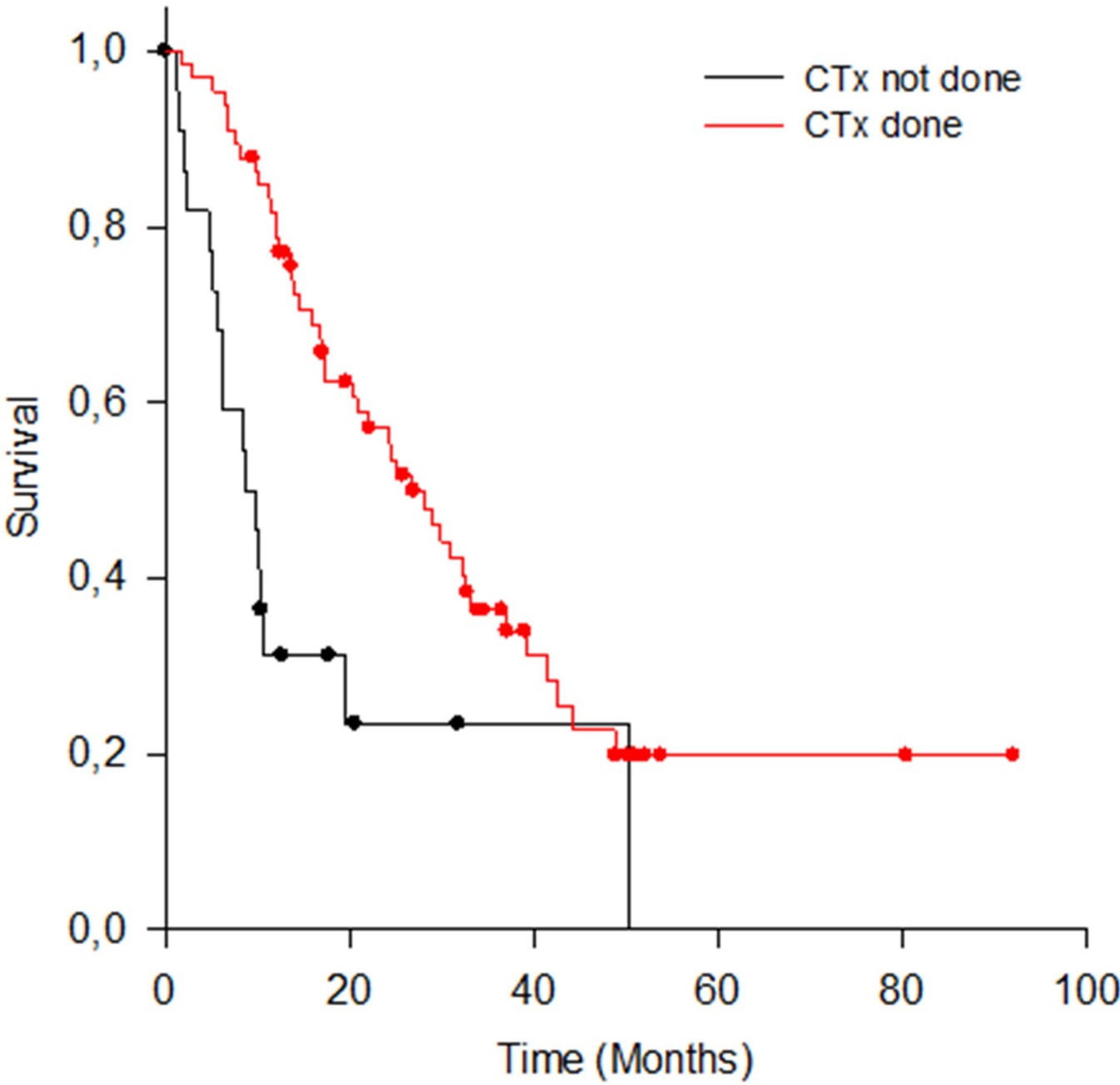

**Fig 6. OS for patients with and without CTx.**

different study results regarding the influence of the resection margin on survival [8, 34–37]. In our study, we compared OS and long-term survival based on the UICC classification (UICC-R0 vs. UICCC-R1), and based on RCP classification or rather the circumferential resection margin (CRM negative vs. CRM positive). Using the RCP R-classification (> 1.0 mm margin clearance) two-thirds (66,3%) of the patients were considered to have incomplete resection margins (i.e. CRM positive). When UICC R-classification (0 mm margin clearance) is applied, almost three-quarters (75,8%) of the patients had complete curative resection margins. According to UICC R-classification we found a significant difference in OS between R0

**Table 2. Results of the multivariate logistic regression to predict long-term survival >36 months.**

| Ind. Variable | Coefficient | SE | Wald Statistic | p-value | Odds Ratio | 5% lower CI | 95% upper CI |
|---|---|---|---|---|---|---|---|
| Constant | -2.185 | 1.070 | 4.169 | 0.041 | 0.112 | 0.0138 | 0.916 |
| LNR | -3.458 | 2.377 | 2.116 | 0.146 | 0.0315 | 0.00030 | 3.324 |
| CRM status | -1.021 | 0.652 | 2.452 | 0.117 | 0.36 | 0.1 | 1.293 |
| syst. Therapy received | 1.908 | 1.086 | 3.087 | 0.079 | 6.742 | 0.802 | 56.676 |

Ind Variable: independent Variable

SE: Standard Error

CI: Confidence Interval Boundary

and R1 resection. In contrast to the findings of Gebauer et al. [38], we showed a significant difference on survival based on R0 versus R1, with median OS of 30 and 16 months, respectively. Thus, our results suggest that the UICC R status performs well as a single prognostic factor for OS. On the other hand, stratification by CRM status only yielded a clinically relevant

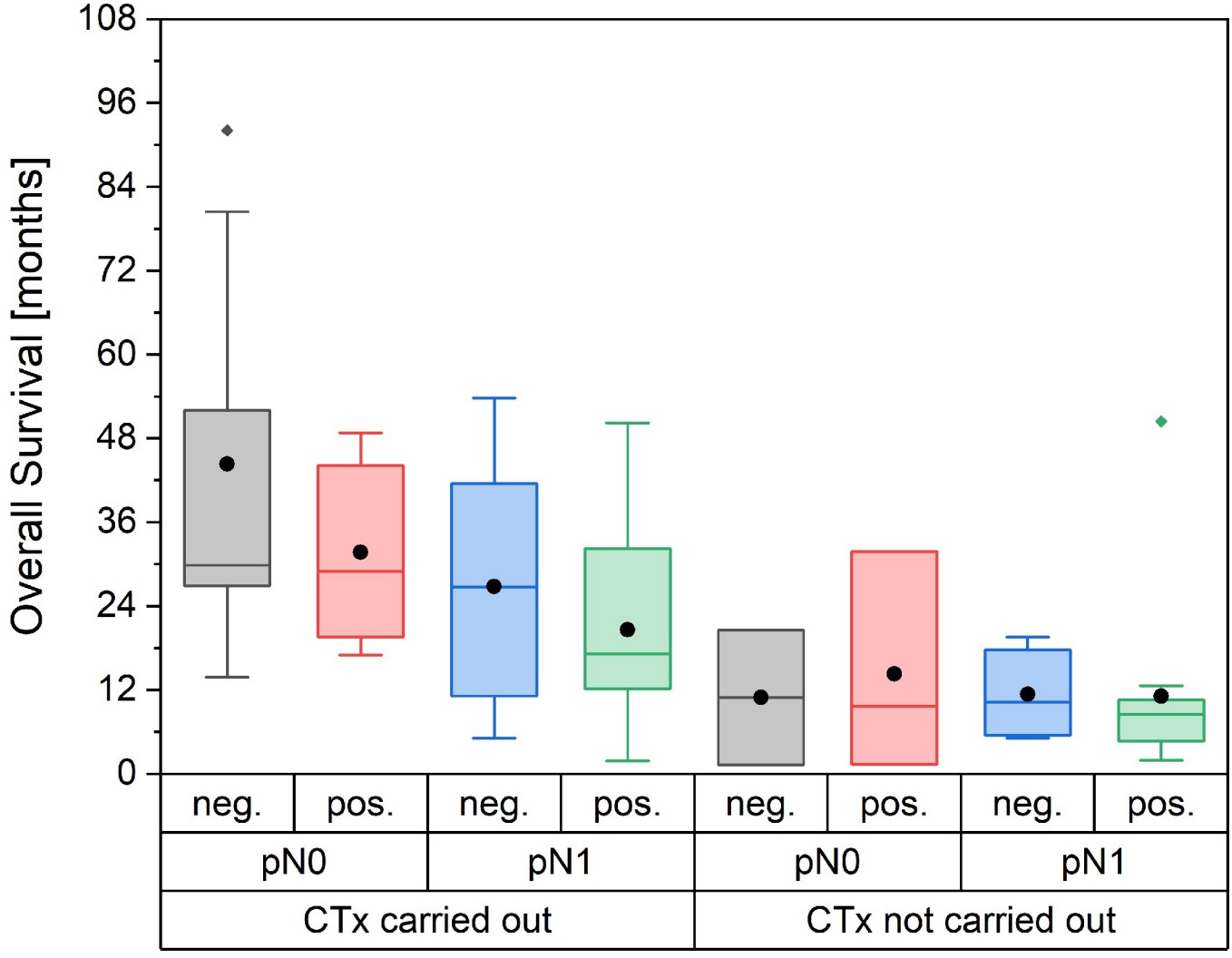

**Fig 7. OS as a function of CTx, PN status, and CRM status.**

difference in OS of more than 12 months (17 vs 30 months, CRM positive vs. CRM negative respectively), which, however, was not statistically significant. Thus, our results are somewhat in line with the results of Campell et al [39] who could show that the survival rate after CRM positive resection is significantly worse than after CRM negative resection in univariate analysis.

However, CRM seems to especially influence long-term survival since the likelihood of surviving 36 months or longer is twice as high in CRM negative patients (20 vs. 10%).

The impact of adjuvant chemotherapy has been discussed in different trials [9–11, 40, 41]. Our study underlines the fact that systemic therapy prolongs OS in patients regardless of whether the resection margin is tumor-free or not. The effectiveness of neoadjuvant therapy on borderline tumors or primarily local advanced tumors and its relevance for long-term survival will have to be evaluated in future studies.

The results of the statistically very robust multifactorial logistic regression suggest that long-term prognosis according to PDAC depends on several factors simultaneously. In our case, margin positivity according to RCP guidelines, chemotherapy and nodal status have the strongest influence on patients´ long-term survival. Out of those, administration of chemotherapy was the most relevant predictor of long-term survival following pancreatic cancer resection, a finding that is consistent with many published studies (9–11, among others). The multivariate analysis results also seem to imply that UICC R classification is no prognostic factor for long-term survival in a multivariate setting. This result, together with the result of the analysis of the relationship between resection margin and overall survival (Fig 3), indicates that the widest possible resection margin could be decisive for long-term survival, and that not only the R status itself is relevant for establishing a valid prognosis.

The reporting of microscopic margin involvement (R1 resection) varies considerably in literature from 20% to 85%. The RCP R0 resection rate is reported in many European centers between 15–30%. In our study, RCP R0 resection (CRM negative) rate was 33.7%, which is comparable to other studies using standardized histopathological reporting as described by Verbeke et al. among others [21, 36, 37, 42]. Interestingly, the recent publications show that the growth pattern of pancreatic cancer is more dispersed than that of rectal cancer [43]. This finding implies that the R0 definition based on RCP guidelines may also underestimate the rate of incomplete resection. However, it remains unclear whether wider margins have an additional benefit on overall or long-term survival as reported in other studies [35, 38, 44]. In our data, we see a tendency towards larger resection margins resulting in increased OS. However, because of the sparse data for really large resection margins (i.e., > 10 mm) there is no valid statement possible. Larger data sets based on standardized histopathological regimen are needed to further analyze this. As a matter of fact, we fully agree with del Carmen Gómez-Mateo et al. [45] that the lack of consensus on margins not only affect their nomenclature and standardized inclusion in the pathological report, but also the definition of R1. A solid tumor with such devastating OS urgently needs international consensus, so that a general overview of the multimodal concepts, like it´s practiced in e.g. esophageal and rectal cancer, can be discussed genuinely.

From a statistical point of view, the multivariate analysis used here has less strict requirements for the distribution function of the dependent data and therefore allows for more reliable statements than the typically employed Cox Proportional Hazards Regression model. Also, this analysis needs smaller samples size to achieve high power. The multivariate model established this way, predicts death or survival longer than 36 months correctly in about 70% of all cases based on the administration of chemotherapy, the nodal status and the CRM status. This is above the predictive power of most of the previous univariate or multivariate models [6, 8, 13, 19, 36]. Apparently, however, one or more influencing variables are still missing to

improve the predictability for long-term survival after PDAC, which consequently shows that some important aspects are not understood or receive too little attention. For instance, Groot et al. [46] demonstrated that most patients with PDAC have systemic disease at the time of resection, thus suggesting a unique biological difference of PDAC leading to different patterns of recurrence. Consequently, we agree with Demir et al. [47] that we have to analyze factors other than margin involvement in order to improve the prognosis of PDAC.

In summary of the results, it can be said that for the purpose of further studies CRM status should indeed be an important part of the histopathological processing. Like a recent study from Strobel et al. [48], we validated the redefined RCP definition currently recommended in Europe, that uses a 1 mm resection margin as cut off, at least with regards to long-term survival.

There are some limitations of our study: 1. In the present study, 5% of patients received neoadjuvant treatment, including radio-chemotherapy; this may have biased the evaluation of resection margins to some extent. 2. In this study we did not analyze the relevance of individual or impact of different circumferential resection margin on survival. More detailed studies on the surgical margin status are essential in the future and lastly 3. This study was limited by its single center design with a relatively small group (n = 95 patients). However, post-hoc power calculation for the multivariate logistic regression confirmed adequate sample size (power > 0.9), and overall tendencies are similar to those published by Strobel et al. [48]. 4. The OS curves displayed here are not yet fully matured because the median follow-up time was 21.8 months with 34% of all data censored. Since OS was mainly used to describe our patient collective in comparison to other studies, and factors of long-term survival were more relevant in our analysis, this is only marginally relevant to the main focus of this study.

However, there are also some advantages compared to previous studies: this study analyzed a standardized, homogenous group of patients for curative intent. The majority of previous studies have included tumors not confined to the head of the pancreas, and/or tumors other than ductal adenocarcinomas [49–52], among others. The study population in our study is very homogeneous and therefore provides valid results for this specific study population. Furthermore, the standardized way of specimen preparation is a great advantage of this study compared to other studies published so far, where there is a huge difference in the reporting of a resection margin status, suggesting inconsistent reporting of histopathological specimens as has already been mentioned. In addition to typically reported OS, we specifically analyzed factors relevant to long-term survival, which is especially important to investigate in a disease with such short life expectancy as PDAC.

## Conclusion

Tumor-free resection margins remain an independent and clinically relevant predictor for survival or prognosis of adenocarcinoma of the pancreatic head. The clinical significance does not depend on which definition for a tumor free resection margin is used.

Our results suggest that the RCP R classification (R0 free margins > 1mm / CRM negative) may be a useful predictor, especially for long-term survival, and indirectly a predictor of the invasive potential of pancreatic cancer compared to the current UICC R0 classification, indicating a disperse growth pattern of pancreatic cancer. However, it is still unclear whether the additional classification according to RCP is a better long-term predictor of local recurrence or distant metastasis. As a matter of fact, our study confirms that chemotherapy is the most significant predictor of long-term survival following pancreatic cancer resection regardless of the margin involved.

To improve predictability and outcome in pancreatic cancer, which is obviously very multifaceted, large randomized prospective studies are required. These studies should put more

focus on the influence of different resection margins (circumferential resection margins) using the Verbeke et al. protocol in order to establish an evidence based standardized reporting of the resection margin.

## Supporting information

**S1 Fig. Comparison of the two types of classification for the resection margin used in this study.**
(TIF)

**S1 File.**
(PDF)

## Acknowledgments

The authors would like to thank Mrs. Kristin Eilermann, Mrs. Sonja Janssen, Mrs. Bianca Sahlmann, and Mr. Fynn Piastowski for the maintenance of the prospective database used here.

## Author Contributions

**Conceptualization:** Dirk Weyhe, Andrea Tannapfel.

**Data curation:** Verena Nicole Uslar, Ingo Stricker.

**Formal analysis:** Verena Nicole Uslar.

**Investigation:** Dennis Obonyo, Ingo Stricker.

**Methodology:** Dirk Weyhe, Dennis Obonyo, Verena Nicole Uslar, Andrea Tannapfel.

**Project administration:** Dirk Weyhe.

**Resources:** Dirk Weyhe, Andrea Tannapfel.

**Supervision:** Dirk Weyhe, Andrea Tannapfel.

**Validation:** Dennis Obonyo.

**Visualization:** Verena Nicole Uslar.

**Writing – original draft:** Dennis Obonyo, Verena Nicole Uslar.

**Writing – review & editing:** Dirk Weyhe, Ingo Stricker, Andrea Tannapfel.

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
