## [Decision Letter · Decision Letter 0]

13 Jan 2021

PONE-D-20-38146

Predictive factors for long-term survival after Surgery for Pancreatic ductal adenocarcinoma: making a case for  standardized reporting of the resection margin  using certified cancer center data

PLOS ONE

Dear Dr. Uslar,

Thank you for submitting your manuscript to PLOS ONE. After careful consideration, we feel that it has merit but does not fully meet PLOS ONE’s publication criteria as it currently stands. Therefore, we invite you to submit a revised version of the manuscript that addresses the points raised during the review process.

We look forward to receiving your revised manuscript.

Kind regards,

Ulrich Wellner, PD Dr. med.

Academic Editor

PLOS ONE

Journal Requirements:

3. We noted in your submission details that a portion of your manuscript may have been presented or published elsewhere.

"preliminary results were published as an abstract in a conference:

Weyhe, D., Uslar, V., Sahlmann, B., & Tannapfel, A. (2018, January). Circumferential Resection Margin influences long term survival after pancreatic adenocarcinoma. In ONCOLOGY RESEARCH AND TREATMENT (Vol. 41, pp. 171-171). ALLSCHWILERSTRASSE 10, CH-4009 BASEL, SWITZERLAND: KARGER."

Please clarify whether this conference proceeding or publication was peer-reviewed and formally published. If this work was previously peer-reviewed and published, in the cover letter please provide the reason that this work does not constitute dual publication and should be included in the current manuscript.

Reviewers' comments:

Reviewer's Responses to Questions

**Comments to the Author**

1. Is the manuscript technically sound, and do the data support the conclusions?

Reviewer #1: Partly

Reviewer #2: Yes

2. Has the statistical analysis been performed appropriately and rigorously? 

Reviewer #1: Yes

Reviewer #2: Yes

3. Have the authors made all data underlying the findings in their manuscript fully available?

Reviewer #1: Yes

Reviewer #2: Yes

4. Is the manuscript presented in an intelligible fashion and written in standard English?

Reviewer #1: Yes

Reviewer #2: Yes

5. Review Comments to the Author

Reviewer #1: Surgical procedures are the most effective treatment for pancreatic ductal adenocarcinoma. Currently, there is no uniform standard for surgical margin management. The authors analyzed and compared CRM status and R status on survival after surgery for PDAC of the pancreatic head. Here are some comments and opinions.

1. the positive surgical margin is related to the aggressiveness of tumor. Is it possible to include relevant data such as tumor size, tumor pathological differentiation, and CA199 level into the study

2. is there a difference in the recurrence-free survival of patients

3. 36 months as the cut-off point for long-term survival, how is this time chosen?

4. is it possible to provide data on the causes of patients' death?

Reviewer #2: The work by Dirk Weyhe et al., entitled “Predictive factors for long-term survival after Surgery for Pancreatic ductal adenocarcinoma: making a case for standardized reporting of the resection margin using certified cancer center data” systematically explored the issue that standardized reporting of the resection margin after PDAC. The authors performed a retrospective study of 95 patients with pancreatic cancer to analyze two different resection margin definitions using univariate OS and multivariate long-term survival. Results showed that chemotherapy, nodal status and resection margin according to UICC R status were univariate factors for OS after PDAC, while long-term survival seems to depend on wider resection margins than those used in UICC R classification. Therefore, authors suggested that standardized histopathological reporting (including resection margin size) should be agreed upon.

Manuscript was well prepared. The topic of this manuscript is meaningful, which focus on the resection margin easily neglected in clinical treatment. Limitations of this study also be discussed. However, several concerns about this manuscript should be pointed out and revised.

1.      Although the reasons for using logistic regression instead of Cox regression was explained, the survival time of patients with pancreatic cancer is still an important observation point. It is recommended to use multivariate Cox regression to analyze the long-term survival and present the results in supplementary table.

2.      It is needed to explain why nodal status and chemotherapy administration, instead of other factors, are included in the main observation indicators and the subsequent multi-factor logistic regression.

3.      The difference in the definition of the resection margin range is the main difference between the two reporting methods (CRM versus UICC R, 1 mm versus 0 mm). In this case, it is recommended to add the analyses of survival of patients with pancreatic ductal adenocarcinoma with a margin of 0 ~ 1 mm, and compare

6. PLOS authors have the option to publish the peer review history of their article (what does this mean?). If published, this will include your full peer review and any attached files.

Reviewer #1: No

Reviewer #2: No

---

## [Author Response · Author response to Decision Letter 0]

22 Feb 2021

Dear reviewers, dear editor,

Thank you very much for your time and effort in reviewing this manuscript. We were happy to note that requests for changes were few, and that the reviews were overall very positive. We addressed all comments and changed the manuscript accordingly. We hope that with the changes made due to your feedback, the revised manuscript is now clearer at those critical points. At least we think that the paper has gained in quality through your comments and the changes based on them.

Below you will find your comments and our respective responses.

With kind regards and thank you very much for your time,

Verena Uslar for all authors

Editors’ comments

 Sorry for this inconvenience. We changed the layout accordingly. 

 The respective paragraph now reads: The study was approved by the medical Committee for Research Ethics at the University of Oldenburg (reference number: 2019-071) without the need for Informed Consent due to the retrospective nature of this study, and was registered with the German Clinical Trials Registry (reference number DRKS0017425). It followed the Helsinki Declaration.

3. We noted in your submission details that a portion of your manuscript may have been presented or published elsewhere.

"preliminary results were published as an abstract in a conference:

Weyhe, D., Uslar, V., Sahlmann, B., & Tannapfel, A. (2018, January). Circumferential Resection Margin influences long term survival after pancreatic adenocarcinoma. In ONCOLOGY RESEARCH AND TREATMENT (Vol. 41, pp. 171-171). ALLSCHWILERSTRASSE 10, CH-4009 BASEL, SWITZERLAND: KARGER."

Please clarify whether this conference proceeding or publication was peer-reviewed and formally published. If this work was previously peer-reviewed and published, in the cover letter please provide the reason that this work does not constitute dual publication and should be included in the current manuscript.

 The abstract submitted to the conference was peer-reviewed, and was then published in the conference proceedings. Since the publication constitutes only of a 300 word abstract, this does not constitute dual publication from our point of view. We hope you agree. 

 We were not sure about the legal ramifications of publishing our data at the time of submission. We have cleared our questions with our ethics committee and will upload our data set as supporting information with the revised manuscript 

Reviewers' comments:

1. Is the manuscript technically sound, and do the data support the conclusions?

Reviewer #1: Partly

Reviewer #2: Yes

 2. Has the statistical analysis been performed appropriately and rigorously? 

Reviewer #1: Yes

Reviewer #2: Yes

3. Have the authors made all data underlying the findings in their manuscript fully available?

Reviewer #1: Yes

Reviewer #2: Yes

 4. Is the manuscript presented in an intelligible fashion and written in standard English?

Reviewer #1: Yes

Reviewer #2: Yes

 5. Review Comments to the Author

Reviewer #1: 

Surgical procedures are the most effective treatment for pancreatic ductal adenocarcinoma. Currently, there is no uniform standard for surgical margin management. The authors analyzed and compared CRM status and R status on survival after surgery for PDAC of the pancreatic head. Here are some comments and opinions.

1. the positive surgical margin is related to the aggressiveness of tumor. Is it possible to include relevant data such as tumor size, tumor pathological differentiation, and CA199 level into the study

 we included tumor size and pathological differentiation in the table with the patient characteristics. We refrained from including CA-19-9 since the time point for which this information was available was very heterogenous across patients, and therefore would not added viable information.

2. is there a difference in the recurrence-free survival of patients

 Yes, RFS differed between CRM pos and CRM neg, and between R0 and R1 respectively. We included this in the patient characteristics table, in the results, and one in sentence in the discussion.

3. 36 months as the cut-off point for long-term survival, how is this time chosen?

 3 yrs were chosen because it corresponds roughly to the time when about 1/3 of all patients were still alive. We deemed it interesting to analyze why this collective had a better long-term survival as compared to the average PDAC patient. We agree this should be mentioned, and therefore added a sentence explaining this in the methods section. 

4. is it possible to provide data on the causes of patients' death?

 sadly, due to the retrospective character of this study, we only know cause of death for about a third of our patients. The tumor documenters who maintain these data unfortunately often receive information on the time of death only from obituaries. However, in those cases we do know, about 90% died because of recurring disease. We included a sentence with regards to this in the manuscript. 

Reviewer #2: 

The work by Dirk Weyhe et al., entitled “Predictive factors for long-term survival after Surgery for Pancreatic ductal adenocarcinoma: making a case for standardized reporting of the resection margin using certified cancer center data” systematically explored the issue that standardized reporting of the resection margin after PDAC. The authors performed a retrospective study of 95 patients with pancreatic cancer to analyze two different resection margin definitions using univariate OS and multivariate long-term survival. Results showed that chemotherapy, nodal status and resection margin according to UICC R status were univariate factors for OS after PDAC, while long-term survival seems to depend on wider resection margins than those used in UICC R classification. Therefore, authors suggested that standardized histopathological reporting (including resection margin size) should be agreed upon.

Manuscript was well prepared. The topic of this manuscript is meaningful, which focus on the resection margin easily neglected in clinical treatment. Limitations of this study also be discussed. However, several concerns about this manuscript should be pointed out and revised.

 1. Although the reasons for using logistic regression instead of Cox regression was explained, the survival time of patients with pancreatic cancer is still an important observation point. It is recommended to use multivariate Cox regression to analyze the long-term survival and present the results in supplementary table.

 This is a very good suggestion that we would normally be very happy to consider. And indeed the results of the Cox regression largely support the results of our logistic regression. However, the Cox regression is not appropriate for our data, as the requirements for the application of this statistical test are violated. We were sorry to note that the manuscript was not very clear in that regard, and changed formulations at a few paragraphs accordingly, to better reflect that decision. In addition to one of the authors being very proficient in statistical analysis, we cleared this with a biometric expert at our university to make sure. 

2. It is needed to explain why nodal status and chemotherapy administration, instead of other factors, are included in the main observation indicators and the subsequent multi-factor logistic regression.

 It is known from previous studies using univariate analyses that nodal status and chemotherapy are two of the most important factors influencing survival after PDAC. Therefore, we included those in the univariate analysis, to establish if our data supports this, which indeed is does. In the multivariate analysis we included the following variables as described in the manuscript: age at the time of surgery, sex, ASA score, pT, pN, pM, lymph node ratio, pR, CRM, and systemic therapy received or not. On the one hand, variable selection was made for pragmatic reasons, since those variables were readily and more importantly reliably available. On the other hand, the included variables all made sense from a clinical point of view, and we think the chosen variables represent the most important patient characteristics which might have confounded our findings. We included the following paragraph in the manuscript to that regard. 

3. The difference in the definition of the resection margin range is the main difference between the two reporting methods (CRM versus UICC R, 1 mm versus 0 mm). In this case, it is recommended to add the analyses of survival of patients with pancreatic ductal adenocarcinoma with a margin of 0 ~ 1 mm, and compare

 Thank you very much for this comment. We are aware of the ongoing discussion on this topic. For example, we have quoted Strobel et al. However, from our point of view, three arguments speak against this approach. Firstly, our resected specimens have been analyzed by our colleagues in pathology since 2010 according to the Verbeke protocol, and our tumor documenters use the 8th Edition of the UICC TNM classification of malignant tumors, 2017, for R classification, so that our analysis also focuses on these classifications. We now added a sentence in the methods section (histopathological assessment) to explain this. Secondly, since these classifications also are still most common today, at least in Germany, the analysis we have chosen is also very relevant for many colleagues with regard to their own data. And thirdly, with some reworking, we could probably make the desired classification and then analyze it. However, according to our current knowledge, there are only 11 patients in the one group, which does not allow for a meaningful analysis. In addition, we would like to point out, that doing this analysis would in our view completely change the intention and the structure of this paper.

---

## [Editor Report · Decision Letter 1]

3 Mar 2021

Predictive factors for long-term survival after surgery for pancreatic ductal adenocarcinoma: making a case for standardized reporting of the resection margin using certified cancer center data

PONE-D-20-38146R1

Dear Dr. Uslar,

We’re pleased to inform you that your manuscript has been judged scientifically suitable for publication and will be formally accepted for publication once it meets all outstanding technical requirements.

Kind regards,

Ulrich Wellner, PD Dr. med.

Academic Editor

PLOS ONE
---

## [Editor Report · Acceptance letter]

9 Mar 2021

PONE-D-20-38146R1 

Predictive factors for long-term survival after surgery for pancreatic ductal adenocarcinoma: making a case for standardized reporting of the resection margin using certified cancer center data 

Dear Dr. Uslar:

I'm pleased to inform you that your manuscript has been deemed suitable for publication in PLOS ONE. Congratulations! Your manuscript is now with our production department. 

Kind regards, 

on behalf of

Dr. Ulrich Wellner 

Academic Editor

PLOS ONE